# Enhancing the Spermidine Synthase-Based Polyamine Biosynthetic Pathway to Boost Rapid Growth in Marine Diatom *Phaeodactylum tricornutum*

**DOI:** 10.3390/biom14030372

**Published:** 2024-03-19

**Authors:** Hung-Yun Lin, Chung-Hsiao Liu, Yong-Ting Kang, Sin-Wei Lin, Hsin-Yun Liu, Chun-Ting Lee, Yu-Chen Liu, Man-Chun Hsu, Ya-Yun Chien, Shao-Ming Hong, Yun-Hsuan Cheng, Bing-You Hsieh, Han-Jia Lin

**Affiliations:** 1Center of Excellence for the Oceans, National Taiwan Ocean University, Keelung 202301, Taiwan; hungyun@mail.ntou.edu.tw; 2Department of Bioscience and Biotechnology, National Taiwan Ocean University, Keelung 202301, Taiwanma2gie3ya@gmail.com (M.-C.H.);

**Keywords:** diatoms, spermidine synthase, activity site, rapid growth

## Abstract

Diatoms, efficient carbon capture organisms, contribute to 20% of global carbon fixation and 40% of ocean primary productivity, garnering significant attention to their growth. Despite their significance, the synthesis mechanism of polyamines (PAs), especially spermidine (Spd), which are crucial for growth in various organisms, remains unexplored in diatoms. This study reveals the vital role of Spd, synthesized through the spermidine synthase (SDS)-based pathway, in the growth of the diatom *Phaeodactylum tricornutum*. PtSDS1 and PtSDS2 in the *P. tricornutum* genome were confirmed as SDS enzymes through enzyme-substrate selectivity assays. Their distinct activities are governed primarily by the Y79 active site. Overexpression of a singular gene revealed that PtSDS1, PtSDS2, and PtSAMDC from the SDS-based synthesis pathway are all situated in the cytoplasm, with no significant impact on PA content or diatom growth. Co-overexpression of PtSDS1 and PtSAMDC proved essential for elevating Spd levels, indicating multifactorial regulation. Elevated Spd content promotes diatom growth, providing a foundation for exploring PA functions and regulation in diatoms.

## 1. Introduction

Polyamines (PAs) comprise a collection of aliphatic compounds, characterized by the presence of multiple amino groups [1]. PAs, found universally in all living organisms, are vital for sustaining fundamental physiological functions [1,2,3]. Among them, spermidine (Spd) is considered indispensable for biological growth and development, exhibiting significant benefits in promoting health, combating aging, and extending lifespan [4,5,6]. Manipulating the duration of inhibitor exposure in vivo emerges as a superior strategy to elucidate the physiological significance of PAs [7]. Previous reports demonstrated that cyclohexylamine (CHA) effectively hinders the synthesis of Spd in cells [7,8,9]. Whether in prokaryotes or eukaryotes, the introduction of CHA results in a notable reduction in intracellular Spd content, thereby significantly impeding the growth of organisms and compromising their resilience in the face of adversity, as observed in *Cryptococcus neoformans* (yeast) [7], *Escherichia coli* (bacteria) [8], and *Strombidium parasulcatum* (ciliate) [9]. These negative physiological effects caused by external addition of CHA can be alleviated through exogenous supplementation of Spd, providing further evidence that Spd is a pivotal biomolecule influencing biological growth [7,8,9].

PAs serve as a significant contributor to the dissolved organic matter (DOM) in the marine environment [10,11]. Dynamic changes in the environmental concentration of PAs are highly related to planktons, with particular emphasis on diatoms and bacterioplankton communities [10,11,12,13]. Diatoms, noteworthy microalgae in the marine environment, exhibit ecological success due to their exceptional efficiency in carbon sequestration and crucial role in driving primary productivity, making them indispensable organisms within the marine ecosystem [14,15]. Many reports indicate that diatoms possess the capability to produce and release PAs, and algal blooms, which are predominantly composed of diatoms, can influence the PA composition in the environment [10,13]. Furthermore, the population densities of certain bacterioplanktons, such as Roseobacter, have been reported to elevate in response to increasing concentrations of PAs in seawater [12]. Recent research reveals that secondary metabolites produced by diatoms play a role in stimulating the growth of specific bacteria, including Spd [16,17,18]. Additionally, bacterioplankton has been demonstrated to release plant hormones and nitrogen sources that support the growth of diatoms, contributing to the establishment of mutualism [19]. These reports suggest that the secretion of PAs through diatoms contributes to the promotion of ocean carbon cycling and the regulation of marine ecosystems.

Spd synthesis is primarily dominated by the spermidine synthase (SDS)-based synthetic pathway in the majority of organisms [1,2,3]. Originating from arginine, it undergoes conversion into ornithine via arginase, followed by the removal of one CO_2_ through ornithine decarboxylase (ODC) to initiate the synthesis of the foundational PA—putrescine (Put) [20]. The substrate for the PA extension reactions, decarboxylated S-adenosylmethionine (dcSAM), is also a derivative of amino acid [21]. It is synthesized from methionine through the enzymatic actions of methionine adenosyltransferase (MAT) and S-adenosylmethionine decarboxylase (SAMDC) [1,2,3]. The elongation reaction of Put is facilitated by SDS, an aminopropyl transferase (APT) that transfers the aminopropyl group from dcSAM to Put, resulting in the formation of Spd [22]. Several genome databases of diatoms are accessible, and the PA synthetic pathway has been predicted [20,23]. Nevertheless, only one diatom SDS in *Thalassiosira pseudonana* (XP_002294888) has been verified for in vitro activity, despite the presence of an additional candidate SDS gene (XP_002287929) in its genome [24]. Indeed, the in vivo investigation of the diatom Spd synthetic pathway has not been thorough to date [20]. Utilizing *Phaeodactylum tricornutum* as a model species, this study investigates enzyme activities and evaluates physiological functions associated with its two SDS candidate genes (Figure 1). This research contributes to a more comprehensive understanding of the role of SDS in the physiological processes of diatoms.

## 2. Materials and Methods

### 2.1. General Chemical Reagents

All experimental materials, including nutrient salts, vitamins, antibiotics and polyamine standards with reagent grade purity, were purchased from Sigma-Aldrich (St. Louis, MO, USA). dcSAM was obtained from Toronto Research Chemicals (Ontario, Canada). Perchloric acid (HClO_4_), toluene, and HPLC grade acetonitrile (ACN) were purchased from J.T. Baker Company (Phillipsburg, NJ, USA). Restriction enzymes and Gibson Assembly^®^ Master Mix were purchased from New England Biolabs (Ipswich, MA, USA). The PCR-related reagents used in the experiments were obtained from BiOptic (Taipei, Taiwan).

### 2.2. Algal Cultivation and Enumeration

*P. tricornutum* (strain CCMP 632), *Chaetoceros muelleri* (strain CCMP 1316), *Isochrysis galbana* (strain CCMP 1323), and *T. pseudonana* (strain CCMP 1335) were purchased from Provasoli-Guillard National Center for Marine Algae and Microbiota (NCMA). All cultured microalgae were cultivated in L1 medium prepared in sterilized seawater and placed in a full-light incubator with a constant temperature of 20°C and an irradiance level of 115 photons m^−2^ s^−1^.

To monitor the proliferation of cells, the microalgae were cultured in advance to the exponential growth phase and then inoculated to ensure the consistency of the health status of the algae in each experiment. Next, 2 × 10^7^ cell-cultured algae were cultured in 200 mL of fresh medium (starting density 10^5^ cell/mL). The algal samples were collected from the culture flasks daily for counting the cell density during the cultivation process. At the same time, the algae samples were also collected on the 4th day (log phase) to analyze the levels of the PAs.

In order to determine the cell density, 1 mL of the algal sample was stained with Lugol’s reagent and placed in a Sedgewick-Rafter counting chamber (Wildlife Supply Company, Yulee, FL, USA) for 5 min. The cell number was determined with an optical microscope (Optiphot-2, Nikon, Tokyo, Japan) at 100× magnification.

### 2.3. Extraction, Modification, and Analysis of PAs from Microalgae

The algal PA extraction method was referenced from our previous study with slight modification [9]. PA standards, including diaminopropane (Dap), Put, cadaverine (Cad), nor-spermidine (NSpd), Spd, nor-spermine (NSpm), and thermospermine (TSpm), were prepared in serial solutions at concentrations of 0.4, 0.8, 1.6, 3.2, and 6.4 μM (Appendix A). A cultured medium containing 2 × 10^8^ algae cells was sampled through centrifugation (3000× *g*, 10 min, 4 °C), followed by removal of the suspension. The pellet was reconstituted in 450 μL of 5% perchloric acid with 5 μM DAH (internal control). Subsequently, algal cells were disrupted using 0.01 mm diameter glass beads (BioSpec, Bartlesville, MA, USA) with a volume of approximately 100 μL in a ball mill (MM 301, Retsch, Germany) for 10 min, operating at a frequency of 30 Hz. Following separation from cellular debris via high-speed centrifugation (16,000× *g*, 5 min, 4 °C), the transfer of 200 μL from the suspension to a new Eppendorf tube constituted the collection of algal PA extracts.

Subsequently, 275 μL of the derivatization reagent was added (with a final concentration of K_2_CO_3_ at 250 mM and dansyl chloride at 10 mM) and kept for continuous shaking at 100 rpm overnight in the dark. Then, 200 μL of 2 M proline was introduced and vortexed at 100 rpm for 1 h in darkness to facilitate the adsorption of free dansyl chloride. The derivatized PAs were further separated using 150 μL of toluene. Following the removal of the liquid-phase solvent using a vacuum dryer, the residue was subsequently re-dissolved in 1 mL of ACN. 

Derivatized PA samples were analyzed via high-performance liquid chromatography (HPLC; Prominence-i LC-2030C Plus, Shimadzu, Kyoto, Japan), employing a C_18_ column (ZORBAX Eclipse XDB-C18, 5 µm, 4.6 × 150 mm) and fluorescence detector (RF-20Axs, Shimadzu; excitation/emission, 340/515). In accordance with a previous study [9], the mobile phase was employed at a flow rate of 1 mL/min, and a gradient process was established between H_2_O and ACN for signal separation in detection. The entire gradient spanned 50 min. From 0–2.5 min, 65% ACN was used. Between 2.5 and 3.0 min, there was a linear increase in ACN to 75% and it was maintained from 3.0 to 15.0 min. At 15.0 min, there was a linear increase to 85% ACN and it was maintained as such from 16.0 to 22.0 min. From 22.0 to 37.0 min, ACN linearly increased to 100% and it was consistently maintained between 37.0 and 47.0 min. At 47.0 min, ACN linearly decreased to 65%. Between 47.1 and 50.0 min, the mobile phase returned to 65% ACN (Appendix A).

### 2.4. Alignment Analysis

The amino acid sequences of spermidine synthase (SDS) and spermine synthase (SMS) used in this study were obtained from the National Center for Biotechnology Information (NCBI) from encompassed organisms such as *Homo sapiens* (HsSDS, NP_003123; HsSMS, NP_001245352), *Arabidopsis thaliana* (AtSDS1 and AtSDS2, NP_973900 and NP_177188; AtSMS, NP_001190527), and *P. tricornutum* (PtSDS1 and PtSDS2, XP_002185179 and XP_002185737). The amino acid sequences were aligned using the ClustalW method, and similarity analysis was conducted with DNASTAR Lasergene software.

### 2.5. 3D Structure Similarity Analysis

The DALI server was utilized for the similarity analysis of the N-terminal domains of PtSDS1 and PtSDS2 against the Dail-PDB25 subset Protein Data Bank. Both predicted models of PtSDS1 and PtSDS2 were downloaded from AlphaFold Protein Structure database (AFDB; Appendix A) [25,26,27]. To generate the query structure, the N-terminal domains of PtSDS1 (residues 14–74; Appendix A) and PtSDS2 (residues 21–102; Appendix A) were extracted.

### 2.6. Plasmid Construction

The nine plasmids used in this study included pGEX-PtSDS1-His, pGEX-PtSDS2-His, pGEX-PtSDS1 (Y79F)-His, pGEX-PtSDS2 (F79Y)-His, pET-myc-PtSDS1-His, pNR-PtSDS1-EGFP, pNR-PtSDS2-EGFP, pNR-PtSAMDC-EGFP, and pNR-PtSAMDC-His. All plasmids were assembled by ligating amplified PCR products with the Gibson Assembly^®^ Cloning Kit (New England Biolabs). The primer sequences used in the PCR amplification reaction are described in Appendix A. PtSDS1 and PtSDS2 were amplified using the specific primers of PtSDS1 or PtSDS2 genes, and *P. tricornutum* gDNA as a template, and then introduced into the *Nde*I/*Xho*I, *Eco*RI/*Xho*I, and *Eco*RI/*Age*I restriction enzyme site intervals of pET-21a (Millipore, Bedford, MA, USA), pGEX-His (GE Healthcare, Chicago, IL, USA), and pNR-EGFP [28] to generate pGEX-PtSDS1-His, pGEX-PtSDS2-His, pGEX-PtSDS1 (Y79F)-His, pGEX-PtSDS2 (F79Y)-His, pET-myc-PtSDS1-His, pNR-PtSDS1-EGFP, and pNR-PtSDS2-EGFP, a total of six plasmids. As for pET-myc-PtSDS1-His, myc tag and PtSDS1 PCR products were imported into the *Nde*I/*Xho*I interval of pET-21a. In addition, the amplicon generated by the PCR reaction using PtSAMDC-specific primers against *P. tricornutum* gDNA was inserted into the *Eco*RI/*Age*I restriction site of pNR-EGFP to yield pNR-PtSAMDC-EGFP. Alternatively, the PtSAMDC amplicon was introduced into the *Eco*RI/*Hind*III restriction site along with the His tag gene to construct the pNR-PtSAMDC-His plasmid.

### 2.7. Recombinant Protein Expression and Purification

The pGEX-PtSDS1-His, pGEX-PtSDS2-His, pGEX-PtSDS1 (Y79F)-His, pGEX-PtSDS2 (F79Y)-His, and pET-myc-PtSDS1-His plasmids were transferred to the competent cells of ArcticExpress (DE3) RIL strain (Agilent Technologies Inc., Santa Clara, CA, USA). Following an overnight incubation at 37 °C, these transgenic bacterial strains were subsequently diluted tenfold with LB medium (a total volume of 250 mL) and cultivated at 30 °C until reaching the logarithmic growth phase (OD_600_ value ranging between 0.4 and 0.6). Isopropyl β-D-1-thiogalactopyranoside (IPTG) was then introduced to attain a final concentration of 200 μM, followed by induction at 14 °C while agitating at 200 rpm for a duration of 16–18 h. 

After the induced cultivation, the LB medium was replaced with 1/10 volume of binding buffer (50 mM NaH_2_PO_4_, 300 mM NaCl, 10 mM imidazole, pH 8) by centrifugation (5000× *g*, 4 °C, 10 min). The bacterial samples were treated with sonication (S4000, Misonix, Farmingdale, NY, USA) to disrupt the cells and then separated through high-speed centrifugation, with the resulting fraction of the suspension solution containing crude protein. Then, 1 mL of Ni-NTA agarose resin (high density nickel resin, BCL-QHNi-100, Qiagen, Hilden, Germany) was used for purifying the target proteins (i.e., GST-PtSDS1-His, GST-PtSDS2-His, myc-PtSDS1-His, GST-PtSDS1 (F79Y)-His, or GST-PtSDS2 (Y79F)-His) as per the purification protocol provided by the original manufacturer. In brief, after resin equilibration with the binding buffer, the crude extract was passed through the column, and the target recombinant protein was subsequently eluted using a wash buffer (binding buffer containing 25 mM imidazole). After verifying the purity of the eluate through sodium dodecyl sulfate polyacrylamide gel electrophoresis (SDS-PAGE), the recombinant proteins were concentrated using an ultra-centrifugal filter (10 kDa cutoff, Millipore) and exchanged with a Tris-HCl buffer (100 mM, pH 7.4). The protein concentration in the recombinant protein samples was determined using a bicinchoninic aid (BCA) assay. Subsequently, the recombinant protein samples were mixed with an equal volume of glycerol and stored at −20 °C.

### 2.8. Aminopropyl Transferase Activity Assay

The in vitro activity assay for spermidine synthase entailed combining 2 µg of recombinant protein with 10 µM Put and 40 µM decarboxylated S-adenosylmethionine (dcSAM) in a total volume of 50 µL. The mixture was then incubated at 37 °C for 30 min, and the reaction was halted by adding 10 µL of 30% perchloric acid. Subsequently, the PA derivatization modification was carried out, followed by HPLC quantification to assess the progress of the catalytic reaction.

### 2.9. Pull-Down Assay of PtSDS1 and PtSDS2

Equimolar quantities of GST-PtSDS2-His recombinant protein were combined with myc-PtSDS1-His or GST-PtSDS1-His recombinant proteins in a 500 mL PBS buffer at 4°C for 18 h. Subsequently, Glutathione-Sepharose 4B beads (GE Healthcare) were introduced, and the mixture was gently agitated at room temperature for 30 min. After that, the beads were washed with PBS to eliminate nonspecifically bound proteins. Finally, the protein samples were washed with Laemmli sample buffer (GE Healthcare) and further analyzed using Western blotting.

Protein samples were resolved by 12% SDS-PAGE and then transferred to a polyvinylidene fluoride (PVDF) membrane, which was further incubated with an anti-Myc antibody (1:5000 in 5% skim milk) and an anti-mice immunoglobulin G (IgG) antibody (1:10,000 in 5% skim milk) as the primary and secondary antibodies, respectively. The signals were visualized using an ECL Plus chemiluminescence reagent (GE Healthcare) under an imaging system (UVP Biospectrum, Upland, CA, USA).

### 2.10. Gene Expression Pattern Analysis under Dark Synchronization

*P. tricornutum* in the exponential growth phase was inoculated at a density of 10^6^ cells/mL into 1 L of fresh culture medium. The culture was then incubated for 24 h of dark synchronization, using an aluminum foil to block out the light source. Following the restoration of lighting conditions, 25 mL samples were collected every 3 h. 

Before RNA extraction, the collected diatom samples were initially centrifuged (3000× *g* for 10 min at 4 °C) and processed in a ball mill (30 Hz for 10 min; MM 301, Retsch) with glass beads (0.01 mm diameter; BioSpec). Total RNA was extracted utilizing the RNeasy Plant Mini Kit (Qiagen) and RNase-free DNase I set (Qiagen), as per the instructions provided by manufacturer. RNA, quantified using nanodrop, was mixed with an equal volume of nucleic acid denaturant (2 × RNA loading dye, New England Biolabs), and heated for 10 min. Subsequently, the RNA quality was analyzed by agarose gel electrophoresis (2%). 

The total RNA (500 ng) was reverse transcribed into cDNA using the RevertAid First Strand cDNA Synthesis Kit (Thermo Fisher Scientific Inc., Wilmington, DE, USA). The reverse transcription process was carried out at 42 °C for 60 min, followed by 70 °C incubation for 5 min. The procedures and conditions for quantitative PCR (QPCR) analysis were in accordance with the methods detailed in our prior research [28]. In each analysis, a 20 μL reaction mixture consisted of the cDNA sample, specific primer pairs (300 nM), and 1 × SYBR Green Master Mix (Applied Biosystems, Foster City, CA, USA). The specific primer pairs designed to detect the mRNA expression levels of PtSDS1, PtSDS2, and PtCycB1 were generated using Primer Express Software (Applied Biosystems, Thermo Fisher Scientific Inc., Foster City, CA, USA), whereas a reference gene, ribosomal protein small subunit 30S (RPS), was obtained from a prior study [29]. All specific primers required for QPCR are listed in Appendix A. The reactions were conducted using an Mx3000P QPCR system (Agilent Technologies, Santa Clara, CA, USA) with the following thermal profile: an initial cycle at 95 °C for 10 min, followed by 40 cycles of denaturation at 95 °C for 15 sec and annealing/extension at 60 °C for 1 min. To confirm the specificity of the QPCR products, a melting temperature analysis was conducted on the same system, ranging from 55 °C to 95 °C.

The mRNA expression level was quantified by calculating the RNA molar ratio between a target gene (X) and a reference gene (R) using the following equation.
(1)$$\log\left(\frac{X_{0}}{R_{0}}\right)=\log\left(\frac{M_{R}}{M_{X}}\right)+\left(\frac{C_{T,X}}{b_{X}}\right)-\left(\frac{C_{T,R}}{b_{R}}\right)$$

The methodology was elaborated in detail previously [25]. In this equation, X_0_ and R_0_ correspond to the cDNA levels of the target and reference genes, respectively. M_X_ and M_R_ denote the molecular weights of X and R, while C_T,X_ and C_T,R_ represent the cycle threshold numbers for target and reference amplicons, respectively. Additionally, b_X_ and b_R_ indicate the slopes of standard curves for target and reference amplicons, respectively. Each standard curve was generated using serially diluted cDNA, which was reverse transcribed from 500 ng of total RNA (Appendix A). The amplification efficiency (E) was calculated as [10^(−1/slope)^ − 1] × 100%.

### 2.11. The Multi-Pulse Electroporation Transformation for P. tricornutum

All operational procedures were based on a previous report with minor adjustments [30]. *P. tricornutum* was pre-cultivated in EASW medium (http://www3.botany.ubc.ca/cccm/NEPCC/esaw.html) until it reached the logarithmic growth phase. Diatom cells were harvested, followed by two washes with a 375 mM sorbitol solution. Subsequently, 2 × 10^8^ cells were combined with 5 µg of linearized transgenic vector and 40 µg of salmon sperm DNA. After a 30-min incubation on ice, the sample was transferred to a 2-mm electroporation cuvette. Diatom electroporation was performed using the electrotransplantation system (Model PA-4000 Advanced PulseAgile, Cyto Pulse Sciences lnc., Glen Burnie, MA, USA), which generated multiple series of square wave pulses, including 8 poring pulses and 5 transferring pulses (Appendix A). The transformed diatoms were promptly transferred to EASW medium and cultured for 24 h under relatively low light conditions. Subsequently, 5 × 10^7^ diatom cells were smeared on 5 EASW agar plates (1.5%) containing 100 μg/mL Zeocin. About 20–100 brownish colonies appeared in 2–3 weeks under incubation at 20 °C with continuous illumination, which were individually transferred into 2 mL EASW medium containing 100 μg/mL Zeocin. After the 7-day screening period, PCR and green fluorescence measurement (excitation/emission 488/509 nm; SynergyMx, Biotek, Winooski, VT, USA) were used to detect surviving cells to verify the presence and expression of the reporter gene (EGFP). Additionally, a fluorescent microscope (Optiphot-2, Nikon, Tokyo, Japan) was used to observe the transgenic diatoms under 400× magnification.

### 2.12. PCR Verification of Transgenic Genes

Diatom genome was extracted using EasyPrep HY Genomic DNA Extraction Kit (Biotools, Taipei, Taiwan) as per the manufacturer’s instructions. Each PCR reaction contained 1 ng of genomic DNA, 500 nM specific primer pairs, and 1 × Gran Turismo PreMix (Ten Giga Bio, Keelung, Taiwan). The primers used to detect EGFP and the reference gene ribosomal protein small subunit 30S (RPS) are listed in Appendix A. The reaction was carried out in a thermal cycler (GeneAtlas G02, ASTEC, Fukuoka, Japan) and the PCR conditions were as follows: 10 min preheating at 95 °C; an amplification process with 32 cycles of 95 °C for 20 sec, 60 °C for 20 sec, and 72 °C for 30 sec; a final cycle of 72 °C for 5 min; then holding at 14 °C. The PCR products were analyzed by gel electrophoresis using an agarose gel (1%).

### 2.13. Diatom Protein Extraction and Analysis

A total of 5 × 10^8^ diatom cells were harvested and centrifuged (3000× *g*, 10 min, 4 °C) to isolate the pellet. The pellet was resuspended in 200 μL of lysis buffer (10 mM 2-mercaptoethanol, 50 mM phosphate buffer, 0.1% Triton X-100, pH 7.3) with added glass beads (0.01 mm diameter; BioSpec) and mixed using a ball mill (30 Hz, 10 min; Retsch). Subsequent high-speed centrifugation (10,000× *g*, 10 min, 4 °C) yielded the crude protein suspension. The protein concentrations were assessed using the BCA reagent (Thermo Fisher) following the manufacturer’s instructions.

The expression level of EGFP, EGFP-fusion protein, or PtSAMDC-His was determined using sodium dodecyl sulfate-polyacrylamide gel electrophoresis (SDS-PAGE) and Western blotting. A total of 25 μg of crude protein was separated by 12% SDS-PAGE, followed by either Coomassie Brilliant Blue staining or transfer onto a PVDF membrane. After treatment with a blocking buffer (5% skim milk), the PVDF membrane underwent sequential incubation with primary antibodies (anti-EGFP/anti-His; 1:5000 in 5% skim milk) and secondary antibodies (anti-rabbit/anti-mouse immunoglobulin antibodies; 1:5000 in 5% skim milk). The ECL signals on PVDF membrane were detected using an ECL reagent (Millipore) with an imaging system.

### 2.14. Statistical Analysis

All experiments and analyses were validated in three biological replicates. The results were represented as the standard deviation (SD) of the mean (mean ± SD). The Student’s *t*-test was used to determine comparative studies of means. Values of *p* < 0.05 indicated statistically significant differences.

## 3. Results

### 3.1. PA Composition in Marine Microalgae and the Significance of Spd in Growth

Besides the two diatom model species, *P. tricornutum* and *T. pseudonana*, we also included *C. muelleri* and *L. galbana* in our selection for the analysis of PA content. These microalgae are widely prevalent in both ecological and aquaculture contexts. The results indicate variations in the PA content and the primary types of components among these microalgae species (Table 1). Spd content in *P. tricornutum*, *T. pseudonana*, *C. muelleri*, and *L. galbana* was found to be 37.0, 52.5, 14.7, and 7.8 fmol/cell, constituting 28.4%, 12.2%, 16.0%, and 4.1% of the total intracellular PAs, respectively.

Cyclohexylamine (CHA), an inhibitor of Spd synthase activity, was introduced into the culture medium, and the growth rates of the four microalgae were monitored over a six-day period. The outcomes revealed that CHA had varying effects on these microalgae. There was no discernible difference in growth rate between *T. pseudonana* and *L. galbana*, even in the presence of 20 μM CHA. However, the growth rates of *P. tricornutum* and *C. muelleri* were significantly hampered with the addition of 5 μM CHA, and this inhibition of growth rate further intensified as the dose increased (Figure 1). In the presence of 20 μM CHA, the growth rate of *C. muelleri* decreased by 33.0% compared to the control group. Similarly, under these conditions, the growth rate of *P. tricornutum* decreased by 41.2%. Among the four microalgae, *P. tricornutum* exhibited the most significant decrease in growth when exposed to CHA.

Analyzing *P. tricornutum* cultured in medium containing 5 μM CHA for 6 days reveals a significant 85.5% reduction in intracellular Spd content, along with an increase in Put levels, indicating a notable inhibition of Spd synthase activity. Notably, the concentration of NSpd also decreased by 55% (Figure 2A). Increasing the CHA concentration to 20 μM not only further reduced Spd and NSpd concentrations to 88.4% and 90.3%, respectively, but also caused a significant reduction in NSpm content. Speculatively, elevated CHA concentrations might non-specifically inhibit various PA synthetases of *P. tricornutum*.

In the medium containing 5 μM CHA, supplementation with 0.5 μM Spd resulted in a substantial increase in intracellular Spd content. Notably, the addition of just 0.5 μM Spd was sufficient to recede intracellular Spd content to levels observed without CHA supplementation (Figure 2B). The growth curve further demonstrated that, following 7 days of cultivation with an additional 0.5 μM Spd, the cell density of *P. tricornutum* was significantly higher than that of the medium containing 5 μM CHA, effectively returning to the levels observed in the control group. This observation underscores the significant role of Spd in the growth of *P. tricornutum*.

### 3.2. Identifying the Enzyme Responsible for Spd Synthesis in P. tricornutum

In the *P. tricornutum* genome, two genes with a spermidine/spermine synthase-like function were predicted, referred to as PtSDS1 (XP_002185179.1) and PtSDS2 (XP_002185737.1). Generally, both spermidine synthase (SDS) and spermine synthase (SMS) possess three significant active sites, Tyr79, Asp104, and Asp173, which correspond to those found in human spermidine synthase [22]. Significantly, in PtSDS2, a noteworthy change occurred at one of the active positions, where Tyr79 is replaced by Phe79 (Figure 3). Exploring the potential influence of these sequence changes on enzymatic activity and substrate selectivity of PtSDS2 demands further investigation. 

In regard to amino acid sequences, PtSDS1 and PtSDS2 exhibit a 48% sequence identity. PtSDS1 displays a 58% sequence identity to human SDS and 26% to human SMS, indicating a closer resemblance to SDS. However, PtSDS1 shares a 50% similarity with Arabidopsis SDS/SMS, with minimal differences between them (Appendix A). AlphaFold/Dali predictive analytics were further employed to compare the N-terminal region, including the substrate binding pocket (Appendix A). The outcomes indicate challenges in distinctly defining PtSDS1 and PtSDS2 as either SDS or SMS models. Therefore, determining the substrate selectivity of SDS/SMS is not a straightforward task based solely on amino acid sequence or protein structure predictive analytics comparison.

### 3.3. Characterization of the Enzyme Activities of PtSDS1 and PtSDS2

The *E. coli* expression system was employed to produce PtSDS1 and PtSDS2 recombinant proteins (Appendix A), followed by evaluating their enzymatic activity with Put and Spd as substrates to determine their SDS or SMS activity. The results revealed that both PtSDS1 and PtSDS2 recombinant proteins effectively utilize Put as a substrate to synthesize Spd, while failing to use Spd as a substrate for Spm synthesis (Table 2). This confirms that both enzymes exhibit SDS activity and lack SMS activity. Furthermore, we conducted tests with other PAs, including Dap, Cad, NSpd, NSpm, Spm, and TSpm, as potential substrates for these two enzymes. Remarkably, PtSDS1 recombinant protein exhibited a relatively low catalytic activity with Cad as a reactant (Table 2).

The activity of PtSDS2 is 155-fold lower than that of PtSDS1 (Table 2), which may be attributed to the substitution of Tyr79 at a crucial active site with Phe79. Consequently, two recombinant proteins with point mutation, PtSDS1-Y79F and PtSDS2-F79Y, were prepared. In comparison to PtSDS2, it was observed that the enzyme activity of PtSDS2-F79Y, where Phe79 was reverted to Tyr79, exhibited a notable improvement of over 10-fold and yielded a weak catalytic activity for Cad (Table 2). However, it still did not reach a level comparable to that of PtSDS1. On the other hand, the activity of PtSDS1-Y79F was significantly 12-fold lower than that of PtSDS1, unexpectedly, the catalytic ability for Cad increased nearly tenfold (Table 2).

The protein interaction between PtSDS1 and PtSDS2, both of which share the same catalytic function, was also confirmed through the GST pulldown assay. The results indicated that GST-PtSDS2 can indeed form a heterodimer with PtSDS1-myc (Appendix A). The in vitro mixing of PtSDS1-myc and GST-PtSDS2 at a 1:1 molar ratio did not result in any significant change in enzyme activity (Appendix A). 

### 3.4. Variations in the mRNA Expression Levels of PtSDS1 and PtSDS2 throughout the Cell Cycle

To ascertain the significance of PtSDS1 and PtSDS2 in the normal physiology of *P. tricornutum* cells, they were subjected to a 24 h light-free induction to initiate dark synchronization, entering the G_0_ phase. Following the reactivation of the light source, the mRNA expression level of *PtCycB1* (XM_00230.8836), a gene known for its variable expression across different cell cycles, was employed to track the progression of the cell cycle [31]. Among these time intervals, the period from 0 to 6 h following the transition from darkness to light corresponds to the G_1_ phase, the period from 7 to 15 h represents the S phase, and the G_2_/M phase occurs between 16 and 18 h (Figure 4AB). The results indicated that *PtSDS1* and *PtSDS2* display similarly high levels on gene expression, but their expression patterns vary significantly. *PtSDS1* maintains consistent expression throughout each cell cycle, whereas *PtSDS2* exhibits high expression levels exclusively during the G_1_ and G_2_/M phases (Figure 4CD).

### 3.5. Cellular Location of PtSDS1 and PtSDS2 in P. tricornutum

To further investigate the physiological roles of PtSDS1 and PtSDS2 in diatoms, we generated diatom strains with overexpressed *PtSDS1* and *PtSDS2* genes, as well as their upstream gene *PtSAMDC*. The *PtSDS1, PtSDS2,* and *PtSAMDC* genes were inserted into the pNR-EGFP vector, generating the pNR-PtSDS1-EGFP, pNR-PtSDS2-EGFP, pNR-SAMDC-EGFP plasmids (Figure 5A). Subsequently, these plasmids were introduced into the genome of *P. tricornutum* using multiple-pulse electroporation and were screened with zeocin antibiotics, resulting in a transformation efficiency of 3 × 10^−8^ and 2.4 × 10^−8^ colonies/cell. The fluorescence intensity of the transgenic diatoms was quantified, and the diatom strain exhibiting the highest fluorescence level in each group was chosen for further experiments. Following the amplification of genomic DNA from each transgenic diatom strain using EGFP-specific primer pairs, PCR results validated the successful introduction of the target plasmids into each strain (Figure 5B). The Western blot results of soluble proteins further confirmed that these genetically modified diatom strains could proficiently express fusion recombinant proteins, such as PtSDS1-EGFP, PtSDS2-EGFP, and PtSAMDC-EGFP (Figure 5C). Fluorescence microscopic analysis for cell location assessment of PtSDS1 and PtSDS2 revealed that the fluorescent signals were distributed within the cytoplasm, indicating the cytoplasmic localization of PtSDS1-EGFP, PtSDS2-EGFP, and PtSAMDC-EGFP in *P. tricornutum* (Figure 5D).

### 3.6. Physiological Evaluation of PtSDS-Overexpressing Diatom Strains

Preliminarily, we conducted measurements of growth rates in wild-type and various transgenic diatom strains, including PtSDS1-EGFP OE strain, PtSDS2-EGFP OE strain, PtSAMDC-EGFP OE strain, and PtSDS1-EGFP/PtSAMDC-His OE strain to investigate the effects of overexpressing PtSDS1 and PtSAMDC on the growth rate of *P. tricornutum*. The growth curve results indicated that the overexpression of single gene (i.e., PtSDS1, PtSDS2, and PtSAMDC) had a limited impact on promoting growth (Figure 6A). However, co-expressing PtSDS1 and PtSAMDC leads to a remarkable 2-fold increase cell numbers in culture (Figure 6B).

Further examination of the PA composition in each transgenic diatom strain revealed that the sole expression of PtSDS1, PtSDS2, or PtSAMDC did not result in remarkable alterations (Figure 6C). However, co-expression of PtSDS1 and PtSAMDC resulted in a substantial 1.8-fold increase in Spd as well as increase in NSpd and NSpm compared to the wild-type diatom. These results indicate that the enhancement in diatom growth was linked to an elevation in cellular Spd levels mediated by PtSDS1 and PtSAMDC.

## 4. Discussion

This work demonstrates the significant role of Spd produced through the SDS-based synthetic pathway in the growth of diatoms. Figure 1 demonstrates varying impacts of CHA on the growth of four microalgae, including *P. tricornutum*, *T. pseudonana*, *C. muelleri*, and *L. galbana*. Notably, the growth of *P. tricornutum* exhibited the most significant response to CHA, followed by *C. muelleri*. In contrast, the growth rates of *T. pseudonana* and *L. galbana* remained relatively unchanged even under high concentrations of CHA (Figure 1). The Spd levels relative to the total cellular PAs among the four microalgae, ranked from highest to lowest, were *P. tricornutum*, *C. muelleri*, *T. pseudonana*, and *L. galbana* (Table 1). This pattern aligns with the trends observed in Figure 1. These data suggest that Spd may only be highly relevant to the growth of some microalgae.

Previous research highlighted the presence of an uncommon PA, homospermidine (HSpd), in certain eukaryotic cells, serving as a homologous analog to Spd. Knocking out the SDS gene in yeast results in growth inhibition due to reduced Spd production, a suppression phenomenon that can be effectively reversed by adding HSpd [32]. This study lends support to the potential for HSpd to physiologically substitute for Spd. In addition, eukaryotic initiation factor 5 A (eiF5a) is an important translation factor involved in eukaryotic protein synthesis, necessitating activation through modification by Spd [33,34]. Notably, in prokaryotic protein synthesis, the translation elongation factor P (EF-P), analogous to eiF5a, is employed, and its synthesis does not rely on Spd modification [35]. Interestingly, microalgae often harbor prokaryotic genes utilizing horizontal gene transfer mechanisms to acquire bacterial genes [36]. In fact, the presence of EF-P in the genome of *T. pseudonana* is expected (PID 5581 in *T. pseudonana* database of JGI). This observation leads us to speculate that EF-P may serve as a substitute for the function of eiF5a in certain microalgae. Consequently, the occurrence of Spd in some microalgae is notably scarce. For Spd-rich microalgae, such as *P. tricornutum* (Table 1), the regulation of PtSDS is essential for physiological processes.

In Figure 3, the sequence alignment highlights a distinct difference in the active site between PtSDS1 and PtSDS2, where Tyr79 is replaced by Phe79. This substitution may lead to reduced PtSDS2 activity due to the absence of the hydroxyl group in the side chain. Previously, the hydroxyl group of Tyr 79 is thought to augment the capability of Asp173 to deprotonate the N1 amine group in Put [22]. This interaction plays a crucial role in enhancing the activity of spermidine synthase. In fact, the activity of PtSDS2 was 155-fold lower than the that of PtSDS1 (Table 2). Further point mutation experiments demonstrated a similar trend. PtSDS1 exhibited approximately 10 times higher activity than PtSDS1-Y79F, while PtSDS2-F79Y displayed a catalytic activity around 10 times higher than PtSDS2. This underscores that the Tyr79 site is critical for the catalytic activity of PtSDS (Table 2). Similar Y79F substitutions are also present in the SDS of other pennate diatoms, like one of the four SDS candidate genes in *Fragilariopsis cylindrus* (OEU17362). Further exploration is warranted in the future to investigate whether similar distinctions exist in other related species. Nevertheless, the Y79F point mutation failed to induce a greater than 100-fold difference in activity between PtSDS1 and PtSDS2. This indicates that, apart from the active site, variations in other substrate binding sites, such as D80H, G156A, and S174T (Figure 3), might also play a role in influencing protein activity to some extent.

PtSDS1 exhibits substrate selectivity by catalyzing Put and Cad, functioning as an atypical spermidine synthase (Table 2). In vitro activity assays conducted on SDS from rats, *E. coli*, and *Thermotoga maritima* revealed similar characteristics [22,37,38]. Notably, in all cases, the *K_m_* values for Cad were significantly higher than those for Put. Moreover, employing CHA to inhibit SDS activity resulted in a substantial elevation of Put, with no impact on Cad (Figure 2). These results imply that under physiological conditions, Cad does not serve as the primary catalytic target for SDS, but rather, Put dominates. Following CHA treatment, Spd, as well as other higher PAs, including NSpd, NSpm, and TSpm, exhibited significant reductions (Figure 2). Upon Spd supplementation, these PAs showed partial restoration, suggesting that NSpd, NSpm, and TSpm might be downstream metabolites influenced by Spd and regulated by other enzymes. Intriguingly, a parallel trend emerged in the PtSDS1/PtSAMDC co-overexpressing strain. In contrast to the wild type, this strain exhibited not only elevated levels of Spd but also a corresponding increase in NSpd and NSpm within the PA composition, providing additional support for the aforementioned argument (Figure 6). These data underscore the significance of SDS in the PA synthetic pathway of diatoms.

Eukaryotes, along with certain viruses, depend on the SDS-based biosynthetic pathway to produce higher-order PAs, such as Spd [1,2,3]. A recent report revealed that cyanobacteria possess an alternative pathway for Spd synthesis known as the carboxyaminopropylagmatine-based pathway, including carboxyaminopropylagmatine dehydrogenase (CAPADH), carboxyaminopropylagmatine decarboxylase (CAPADC), and aminopropylagmatine ureohydrolase (APAUH), to convert arginine into Spd [39]. Homologs of these enzymes do not exist in diatoms, suggesting that they are more similar to eukaryotic regulatory mechanisms [20,39]. This study reveals that individual expressions of PtSDS1, PtSDS2, and PtSAMDC in *P. tricornutum* do not alter PA composition or cell growth (Figure 6). However, co-expression of PtSDS1 and SAMDC leads to changes in PA composition and an increased growth rate. This observation highlights that, in the SDS-based biosynthetic pathway, the cellular Spd content may be constrained by dcSAM. Similar regulation exists in other microalgae [20]. For instance, the inhibition of SAMDC activity using methylglyoxal bis-guanylhydrazone (MGBG) in *Chattonella antiqua* markedly constrained both cellular Spd content and cell growth [40]. Notably, alongside PtSAMDC, *P. tricornutum* also harbors three SAMDC-APT fusion genes [23]. This suggests the potential existence of additional regulatory mechanisms in diatoms for the synthesis of distinct PAs.

PtSDS1 and PtSDS2, both found in the cytoplasm in *P. tricornutum*, exhibit distinct regulatory influences on the cell cycle (Figure 5). The expression of the *PtSDS1* gene remained consistently stable during cell cycle, whereas the *PtSDS2* gene exhibited an increase during the G_1_ phase and a subsequent decrease during the S phase. In various organisms, Spd has been observed to fulfill a physiological function by stabilizing and safeguarding DNA [1,2]. This regulatory mechanism may aid diatom cells in preparing for DNA replication during the G_1_ phase [41]. The lack of a notable impact on activity when PtSDS1 and PtSDS2 were combined (Appendix A) suggests that PtSDS2 may harbor additional physiological functions. This could involve the formation of metabolite channeling with other functionally correlated catalytic enzymes, a phenomenon referred to as metabolome [42]. Forming a metabolome offers several advantages, including enhanced efficiency in utilizing substrates by enzymes, mitigation of harmful intermediate accumulation, minimized influence of surrounding molecules on metabolic flux, and prevention of substrate or intermediate competition by other metabolic pathways [43]. The presence of a metabolome in the TCA cycle, Calvin cycle, and urea cycle facilitates the more efficient execution of catalytic reactions and resource allocation within the intricate intracellular environment [44,45,46]. This, in turn, enhances an organism’s capacity to adapt to environmental changes [43]. Considering the prediction that at least eight enzymes within the genome of *P. tricornutum* are implicated in the synthesis of distinct PAs [23], our hypothesis is that these enzymes are likely to establish a metabolome or other complex relationship. An essential focus for future research involves investigating whether these proteins can form a metabolon through protein-protein interactions (PPI), potentially contributing to the formation of biologically relevant PAs with more intricate structures, such as long-chain polyamines (LCPAs) [20,23,47].

## Data Availability

All data supporting reported results can be found in the Appendix A.

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
