# Peer review of "Enhancing the Spermidine Synthase-Based Polyamine Biosynthetic Pathway to Boost Rapid Growth in Marine Diatom Phaeodactylum tricornutum"

_biomolecules, 2024, doi:10.3390/biom14030372_

Round 1

Reviewer 1 Report

Comments and Suggestions for Authors

The flow of the manuscript is not good as there is no continuity between the methods and the results. The authors should work on making the manuscript more understandable to the readers. No statistics tests were mentioned in the method sections although I see stats applied in some figures. The English needs improvement as the authors used present tense in some instances, and past in other times.

Comments on the Quality of English Language

Extensive editing of the English language is required.

Author Response

Reviewer #1:

  1. The flow of the manuscript is not good as there is no continuity between the methods and the results. The authors should work on making the manuscript more understandable to the readers.

Response: We appreciate the reviewer's suggestion. We have made adjustments to the sequence of the Materials and Methods section to align it with the content in the results.

  1. No statistics tests were mentioned in the method sections although I see stats applied in some figures.

Response: We have followed the reviewers' suggestion and added a section on statistical methods.

  1. The English needs improvement as the authors used present tense in some instances, and past in other times.

Response: We are grateful to the reviewer for noticing our oversight. Our manuscript has been reviewed and revised by a native speaker and highlight in yellow. Regarding the facilitator, Dr. Binesh Unnikrishnan, we mention his contribution in the Acknowledgments.

Reviewer 2 Report

Comments and Suggestions for Authors

The authors have presented a manuscript outlining the regulation of spermidine synthase to enhance polyamine (PAs) production and promote diatom growth. While the role of PAs in stress physiology, growth, and developmental biology has been extensively studied in higher plants, there is limited exploration regarding their function, biosynthesis pathways, and regulation in both micro and macroalgae.

This meticulously conceptualized and executed work is a well-crafted manuscript that I recently reviewed over the past two months. I found it to be engaging and well-written, with each experiment thoughtfully planned based on the obtained results, demonstrating a comprehensive understanding of the subject matter.

I thoroughly enjoyed reading this manuscript and commend the authors for their thorough research. However, I have a minor suggestion to include a biochemical dataset on different levels of spermidine (Spd) in the cell. While the total Spd levels are measured, it would be valuable to include data on free, soluble, conjugated, and bound insoluble Spd levels in response to CHA inhibitor. Different forms of PAs have distinct functions in a plant cell, and understanding their variations could be linked to the growth of algal cells.

With this minor suggestion, I highly recommend this manuscript for publication.

Author Response

  1. I have a minor suggestion to include a biochemical dataset on different levels of spermidine (Spd) in the cell. While the total Spd levels are measured, it would be valuable to include data on free, soluble, conjugated, and bound insoluble Spd levels in response to CHA inhibitor. Different forms of PAs have distinct functions in a plant cell, and understanding their variations could be linked to the growth of algal cells. With this minor suggestion, I highly recommend this manuscript for publication.

Response: We fully concur with the reviewer's perspective that various polyamine types (free, conjugated, and bound) likely exhibit distinct physiological functions in higher plants. However, in numerous microalgae, spermidine (Spd) predominantly exists in the free form,  showinga concentration difference exceeding a hundred times compared to the bound form Spd [1-3]. Consequently, we contend that focusing solely on the analysis of intracellular free-form Spd in this study should adequately capture relevant cellular physiological phenomena.

  1. Kotzabasis, K.; Senger, H. Free, Conjugated and bound polyamines during the cell cycle in synchronized cultures of Scenedesmus obliquus. Naturforsch. C 1993, 49, 181–185.
  2. Tassoni, A.; Awad, N.; Griffiths, G. Effect of ornithine decarboxylase and norspermidine in modulating cell division in the green alga Chlamydomonas reinhardtii. Physiol. Biochem. 2018, 123, 125–131.
  3. Lin, H.-Y.; Lin, H.-J. Polyamines in microalgae: something borrowed, something new. Drugs 2019, 17, 1.

Round 2

Reviewer 1 Report

Comments and Suggestions for Authors

The manuscript was revised satisfactorily.